# Electroluminescence and hyperphosphorescence from stable blue Ir(III) carbene complexes with suppressed efficiency roll-off

Jie Yan [1,6], Dong-Ying Zhou [2,6], Liang-Sheng Liao [2] ✉, Martin Kuhn [3], Xiuwen Zhou [3] ✉, Shek-Man Yiu[4] & Yun Chi [1,4,5] ✉

Efficient Förster energy transfer from a phosphorescent sensitizer to a thermally activated delayed fluorescent terminal emitter constitutes a potential solution for achieving superb blue emissive organic light-emitting diodes, which are urgently needed for high-performance displays. Herein, we report the design of four Ir(III) metal complexes, f-ct1a – d, that exhibit efficient true-blue emissions and fast radiative decay lifetimes. More importantly, they also undergo facile isomerization in the presence of catalysts (sodium acetate and p-toluenesulfonic acid) at elevated temperature and, hence, allow for the mass production of either emitter without decomposition. In this work, the resulting hyper-OLED exhibits a true-blue color (Commission Internationale de l'Eclairage coordinate $CIE_y = 0.11$), a full width at half maximum of 18 nm, a maximum external quantum efficiency of 35.5% and a high external quantum efficiency 20.3% at 5000 cd m$^{-2}$, paving the way for innovative blue OLED technology.

The performance of organic light-emitting diodes (OLEDs) has steadily improved over the past two decades. However, deep-blue emitters and corresponding blue OLED devices with adequate durability and lifespan are still not available to meet industrial standards, presenting a substantial challenge to future development. In principle, there are two distinctive classes of potentially suitable blue emitters: transition-metal phosphors and organic thermally activated delay fluorescence (TADF) emitters. In recent years, both blue emitters have gained attention because of the diverse designs that could allow for efficient singlet-to-triplet intersystem crossing (ISC) for phosphorescence or a fast reversed process (rISC) for TADF emission. To this end, blue phosphorescent metal complexes, including d$^{10}$-metal Cu(I)[1], Au(I)[2], d$^8$-metal Pt(II)[3,4], Au(III)[5,6], and d$^6$-metal Ir(III) complexes, have been intensively investigated[7–9]. Despite many successful reports, from the viewpoint of thermodynamics, first-row Cu(I) complexes possess relatively weak metal–ligand bonding, while isoelectronic third-row Au(I) complexes carry two monodentate chelates. Both structural motifs cannot endure the extreme stresses that occur during the continuous excitation to produce blue emission, unlike other metal complexes with bidentate or multidentate chelates. Alternatively, both square planar Pt(II) and Au(III) complexes are expected to yield less efficient spin–orbit coupling and slower radiative transition at the excited state in reference to Ir(III) complexes with quasi-octahedral geometry[10]. This results from a reduced zero-field splitting of emitting triplet states for the d$^8$–metal complexes; as a result, Ir(III) emitters are the most suitable candidates (among all transition-metal emitters) for

[1]Department of Materials Science and Engineering, City University of Hong Kong, 999077 Hong Kong, SAR, China. [2]Institute of Functional Nano and Soft Materials (FUNSOM), Jiangsu Key Laboratory for Carbon-Based Functional Materials & Devices, Soochow University, 215123 Suzhou, China. [3]School of Mathematics and Physics, The University of Queensland, Brisbane, Queensland 4072, Australia. [4]Department of Chemistry, City University of Hong Kong, 999077 Hong Kong, SAR, China. [5]Center of Super-Diamond and Advanced Films (COSDAF), City University of Hong Kong, 999077 Hong Kong, SAR, China. [6]These authors contributed equally: Jie Yan, Dong-Ying Zhou. ✉e-mail: lsliao@suda.edu.cn; x.zhou6@uq.edu.au; yunchi@cityu.edu.hk

making durable blue OLED devices. In particular, there is an inverse cubic law relationship between device stability and the triplet exciton lifetime[11]. Here, we demonstrate that efficient and stable blue emitters can be effectively realized using an Ir(III) core framework comprising electron-deficient carbene chelates with asymmetric N-aryl substituents.

## Results

### Synthesis and characterization

To achieve adequately stable blue emitters, we turned to a class of di-N-aryl substituted imidazo[4,5-b]pyrazin-2-ylidene chelates, in which the molecular designs are related to predecessors, such as 5-(*tert*-butyl)-imidazo[4,5-b]pyrazin-2-ylidene[12] and functional purin-8-ylidene[13,14] and imidazo[4,5-b]pyridin-2-ylidene[15,16], except for the presence of two distinctive N-aryl groups. The advantages of dual N-aryl substituents are evident in the superb photophysical properties of the Ir(III) emitter *f*-Ir(cb)₃[12] and its utility in the fabrication of many efficient blue OLEDs[17–19]. Moreover, Thompson and coworkers also confirmed the high stability of an analog *f*-Ir(tpz)₃ against chelate substitution under harsh thermal aging and photolysis conditions[20]. Third, the Pt(II) phosphor PtON-TBBI (BD-2), also with a dual N-aryl-substituted carbene, is capable of producing a blue OLED with an $LT_{70}$ lifetime of 1113 h at 1000 cd m$^{-2}$[21]. These observations are in sharp contrast to those of the N-alkyl-substituted imidazo[4,5-b]pyrazin-2-ylidene chelates, which have obviously exhibited poor stability[20,22]. Hence, the outstanding pro-chelate ct1H₂·OTf and corresponding homoleptic Ir(III) complexes were synthesized, with the expectation that its di-N-aryl substituents will offer better performance, which is consistent with results in the literature[23–25].

Next, the pro-chelate ct1H₂·OTf was synthesized using a multistep protocol (cf., Supplementary Fig. 1 of SI). Subsequently, this pro-chelate was treated with *mer*-trichloridotris(tetrahydrothiophene-κS)iridium(III) (*mer*-IrCl₃(tht)₃) and sodium acetate (NaOAc) in *o*-dichlorobenzene at 155 °C, affording a mixture of four Ir(III) complexes, namely, f-ct1a–d in an approximate ratio of 7:26:33:9, and their structures are depicted in Fig. 1. All products were separated by silica gel column chromatography, followed by recrystallization. Single crystal X-ray diffraction studies were performed to verify the structures. As seen in Supplementary Data 1–4 and Supplementary Figs. 2–5 of the supporting information, these Ir(III) complexes possess quasi-octahedral structures with facially arranged chelates, in which all Ir–C$_{(carbene)}$ distances (2.028–2.050 Å) are notably shorter than Ir–C$_{(aryl)}$ distances (2.077–2.111 Å), showing a stronger Ir–C (dative) interaction for the carbene units[20,22]. Moreover, symmetrical f-ct1a with three p-*tert*-butylphenyl cyclometalates exhibited a relatively

shortened average Ir–C$_{(aryl)}$ distance (2.090(2) Å) and Ir–C$_{(carbene)}$ distance (2.030(2) Å) in comparison to the corresponding Ir–C$_{(aryl)}$ distance (2.104(6) Å) and Ir–C$_{(carbene)}$ distance (2.050(6) Å) of symmetrical f-ct1d with three phenyl cyclometalates. This variation of Ir–C distances reflected the influence of the *tert*-butyl group on the aryl cyclometalate, in which the better electron-donating aryl cyclometalates improved back π-bonding to the accompanying carbene entities.

### Photophysical properties and theoretical studies

The absorption and emission spectra were recorded in toluene at room temperature (RT), and the respective data are presented in Supplementary Fig. 6 and Table 1. All samples exhibited an intense absorption band at ~400 nm, which is characteristic of facial isomers. The emission band is structureless due to the high metal-to-ligand charge transfer (MLCT) contribution. Additionally, it follows a descending order of 468 nm (f-ct1a) ~468 nm (f-ct1b) > 464 nm (f-ct1c) > 462 nm (f-ct1d), in agreement with the numbering of p-*tert*-butylphenyl cyclometalates. Finally, these Ir(III) complexes, except for f-ct1d with a long radiative lifetime of 2573 ns, exhibit a high PLQY of 75–89% and a shortened radiative lifetime of 636–908 ns, offering desirable properties for adequate OLED performance.

The lowest singlet ($S_1$) and lowest triplet ($T_1$) excited states for these Ir(III) complexes were investigated by time-dependent density functional theory (TD-DFT) calculations[26] to gain an understanding of how ligand modifications affected their photophysical properties. The calculated excitation energies to reach the $T_1$ state from the ground state (Supplementary Data 5) are 2.73 eV (453.6 nm), 2.71 eV (458.2 nm), 2.74 eV (451.9 nm), and 2.75 eV (451.6 nm) for f-ct1a–d, respectively (cf. Supplementary Table 1). Hence, the variation in emission energy is less than 0.04 eV (~0.09 kcal mol$^{-1}$), and f-ct1c and f-ct1d achieve slightly greater emission energy than the other two complexes (f-ct1a and f-ct1b) that have indistinguishable experimental phosphorescent peaks. We applied natural transition orbital (NTO) analysis[27] to express the $T_1$ state as a single pair of orbitals. The predominant NTO pairs found for the $T_1$ state (cf. Fig. 1) show f-ct1a–c with a mixture of MLCT, ligand-centered (LC), intra-ligand charge transfer (ILCT), and ligand-to-ligand charge transfer (LLCT) characteristics, whereas f-ct1d exhibits no obvious LLCT contribution. Increased orbital overlap between the NTO pair for the $T_1$ state of f-ct1c or f-ct1d was noted due to their stronger LC characteristics, while the MLCT percentage in the $T_1$ state varied from f-ct1a (20.7%) to f-ct1b (21.5%), f-ct1c (21.5%) and f-ct1d (21.2%) (cf. Supplementary Table 1). Their interplay leads to an enhanced radiative rate constant ($k_r$) for f-ct1c, as a higher MLCT percentage in the $T_1$ excited states should give a faster $k_r$ due to the direct involvement of metal atoms that facilitate

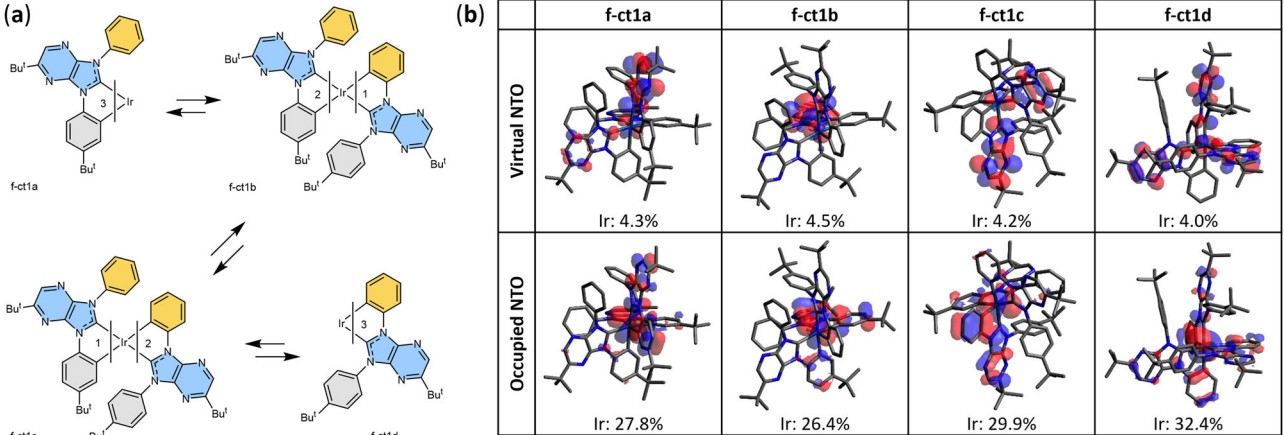

**Fig. 1 | Molecular structures and electronic orbitals of the studied Ir(III) complexes. a** Schematic drawing of molecules. **b** Virtual and occupied natural transition orbitals (NTOs) showing the electron distribution of the lowest triplet ($T_1$) excited states; the contribution of Ir(III) metal to the NTOs is provided.

**Table 1 | Photophysical data of the studied di-N-aryl-substituted imidazo[4,5-b]pyrazin-2-ylidene based Ir(III) complexes at RT**

| | abs $\lambda_{max}$[a] (nm) | em $\lambda_{max}$ (nm) | FWHM[b] (nm) | Φ (%)[c] | $\tau_{obs}$[c,d] (ns) | $\tau_{rad}$[c,e] (ns) | $k_r$ ($10^6$ s$^{-1}$)[e] | $k_{nr}$ ($10^5$ s$^{-1}$)[e] |
|---|---|---|---|---|---|---|---|---|
| f-ct1a | 354 (2.52), 390 (2.25) | 468 | 62 | 80 | 722 | 902 | 1.1 | 2.8 |
| f-ct1b | 348 (2.71), 390 (2.30) | 468 | 63 | 75 | 681 | 908 | 1.1 | 3.7 |
| f-ct1c | 344 (2.81), 390 (2.31) | 464 | 58 | 83 | 528 | 636 | 1.6 | 3.2 |
| f-ct1d | 339 (2.91), 391 (2.32) | 462 | 56 | 89 | 2290 | 2573 | 0.39 | 0.48 |

[a]Those were recorded at a conc. of $10^{-5}$ M in toluene at RT; extinction coefficient (ε) is given in parentheses with a unit of $10^4$ M$^{-1}$ cm$^{-1}$.

[b]Full width at half maximum.

[c]Coumarin 102 (C102) in methanol (Q.Y. = 87% and $\lambda_{max}$ = 480 nm) was employed as standard, and data recorded in degassed toluene solution in $10^{-5}$ M at RT.

[d]Observed lifetimes were obtained from transient PL measurement.

[e]$\tau_{rad} = \tau_{obs}/PLQY$, $k_r = PLQY/\tau_{obs}$ and $k_{nr} = (1-PLQY)/\tau_{obs}$.

phosphorescence, and the increasing orbital overlap also results in an increased $k_r$ due to an increased transition dipole moment[28]. It is also noted that f-ct1d achieves the highest PLQY among all four isomers with both lowered $k_r$ and $k_{nr}$. This might be due to its relatively enhanced structural stability leading to a relatively higher activation barrier from its emissive state to the non-emissive metal-centered states and thus a lower $k_{nr}$[29].

## Catalytic interconversion in solution

The chemistry of these Ir(III) emitters f-ct1a–d is remarkable. After the introduction of a *tert*-butyl group to one N-aryl group of Ir(cb)₃, in giving isomeric f-ct1a–d, we can solve the problem of product separation of Ir(cb)₃ analogs and confirm the fast equilibration and interconversion among all available isomers f-ct1a–d in presence of catalysts at a higher temperature. Moreover, the isomer f-ct1a was obtained in a higher ratio than f-ct1b and f-ct1c in *tert*-butylbenzene solution at 140 °C. Upon changing the solvent to *o*-dichlorobenzene (b.p. = 180 °C), we observed a significant reduction in f-ct1a, together with the formation of a higher proportion of the other three complexes, i.e., f-ct1b–d. Further switching the reaction medium to 1,2,4-trichlorobenzene (b.p. = 214 °C) diminished the yield of f-ct1a, while f-ct1c became the major product, followed by f-ct1b and f-ct1d, in a ratio of 8:6:1. Interestingly, their combined yield remained ~75%, but their relative ratio was notably affected by the applied temperature. Moreover, extensive heating of either Ir(III) emitter in refluxing 1,2,4-trichlorobenzene resulted in neither decomposition nor isomerization, even with the addition of some NaOAc. In contrast, heating of f-ct1a–c in the presence of excess NaOAc and a small amount of *p*-toluenesulfonic acid (TsOH) in refluxing 1,2,4-trichlorobenzene for 36 h afforded an approximately 5:6 mixture of f-ct1b and f-ct1c in high yields, together with trace amounts of f-ct1a and f-ct1d. Furthermore, f-ct1d underwent slower isomerization under similar conditions but yielded more f-ct1b and f-ct1c upon prolonged heating, suggesting greater thermodynamic stability of f-ct1b and f-ct1c with respect to their symmetrical counterparts f-ct1a and f-ct1d. These results pointed to a dynamic equilibration among Ir(III) complexes, in which f-ct1b and f-ct1c are preferable products. Without a doubt, the NaOAc + TsOH catalyzed isomerization proceeded via initial protonation of one aryl cyclometalate, followed by reversible and competitive C–H activation involving both N-aryl groups[30–32]. Accordingly, mass production of f-ct1b or f-ct1c can be realized by repeating the catalytic conversion, followed by chromatographic separation and recycling of undesirable products.

## Electroluminescence (EL) and hyperphosphorescence

Owing to their promising thermal stability (Supplementary Fig. 7), electrochemical (Supplementary Table 3 and Supplementary Fig. 8) and photophysical properties (Table 1), the phosphors f-ct1a–d were utilized in the fabrication of OLED devices with the following architecture: ITO/HAT-CN (10 nm)/TAPC (40 nm)/TCTA (10 nm)/mCP (10 nm)/7–21 wt% dopant in mCBP (20 nm)/TmPyPB (40 nm)/Liq (2 nm)/Al (100 nm). Specifically, HAT-CN (dipyrazino[2,3-f:2',3'-h]

quinoxaline-2,3,6,7,10,11-hexacarbonitrile) is used as the hole injection layer. TAPC (1,1-bis((di-4-tolylamino)phenyl)cyclohexane), TCTA (4,4',4''-*tris*(carbazol-9-yl)-triphenylamine), and mCP (1,3-di(9*H*-carbazol-9-yl)benzene) served as the hole-transporting layers. mCBP (3,3'-di(9H-carbazol-9-yl)-1,1'-biphenyl), TmPyPB (1,3,5-tris(3-pyridyl-3-phenyl)benzene), and Liq (8-hydroxyquinolinolato-lithium) acted as the host material, electron-transporting layer, and electron-injection layer, respectively. The energy levels and molecular structures of the as-mentioned materials and schematic structures of OLEDs are depicted in Supplementary Figs. 9, 10, and the emitting layer (EML) was composed of different hosts and f-ct1a–d in different doping concentrations.

As shown in both Fig. 2a and Table 2, all devices presented blue emissions with peak wavelengths spanning a narrow range from 472 to 476 nm. The EL peak wavelengths of f-ct1c and f-ct1d are blue-shifted in reference to those of f-ct1a and f-ct1b, while all EL peaks are found to be red-shifted by ~8 nm relative to the corresponding photoluminescence (PL) spectra (Supplementary Fig. 11), which could be attributed to the change of polarity of solvent and host matrix and the optical microcavity effects within OLED devices (Supplementary Figs. 12–15). Moreover, their EL spectra showed minimal broadening in comparison to the PL spectra and little variation in bandwidth with varied doping concentrations (Supplementary Figs. 12–15), confirming the suppressed aggregation of emitters. This may result from the greater steric hindrance from the multiple *tert*-butyl groups, which have the capability to reduce triplet–triplet annihilation (TTA) and efficiency roll-off at higher driving current densities[33].

Figure 2b depicts the current density–voltage–luminance (*J–V–L*) characteristics of devices with f-ct1a–d at a doping concentration of 21 wt%. Notably, the f-ct1c and f-ct1d devices show lower turn-on voltages than the f-ct1a and f-ct1b devices. Considering that hole traps are commonly generated in a system with a large gap between the host and dopant, the lowered voltages for f-ct1c and f-ct1d indicate a small mismatch in the HOMO energies between the dopant and host. This result is consistent with the CV data, in which f-ct1c and f-ct1d show anodic shifted oxidation potentials compared to f-ct1a and f-ct1b. To verify the influence of f-ct1c, hole-only and electron-only devices with different doping concentrations of f-ct1c were fabricated (Supplementary Fig. 16). As shown in Supplementary Figs. 17, 18, the hole current is markedly reduced at higher doping concentrations. In contrast, a higher dopant concentration also leads to a decrease in the electron current density, suggesting that the phosphor f-ct1c has a shallow LUMO energy level which also affects the electron transport properties.

Regarding the EL efficiency, the f-ct1c device delivered the highest maximum external quantum efficiency (EQE_max) of 20.0%, whereas the EQE_max values for f-ct1a, f-ct1b, and f-ct1d were 14.3%, 14.4%, and 11.6%, respectively. The high EL efficiency of f-ct1c could be attributed to its fast radiative decay rate (Supplementary Fig. 19) and well-matched HOMO energy level. Compared to f-ct1c, symmetric f-ct1a showed a relatively low efficiency despite its fastest radiative decay rate, which may be attributed to the increased HOMO gap between the dopant and

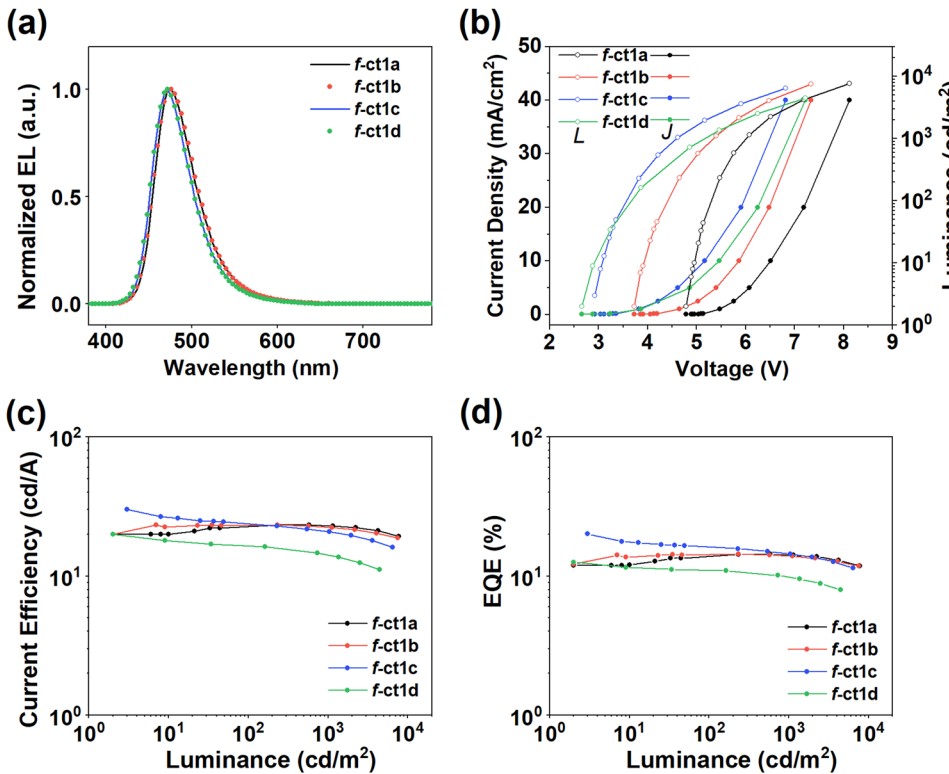

**Fig. 2 | Characteristics of the OLEDs using phosphors f-ct1a–d. a** Electroluminescence spectra. **b** Current density–voltage–luminance ($J$–$V$–$L$) characteristics. **c** Current efficiency–luminance characteristics. **d** EQE–luminance characteristics of devices using f-ct1a–d at a doping concentration of 21 wt%.

**Table 2 | EL parameters of OLEDs with different phosphors (21 wt%) and phosphor-sensitized v-DABNA**

| Dopant | $V_{on}$ (V) | EQE (%)[a] | CE (cd/A)[a] | CIE $(x,y)$[b] | FWHM (nm) | EL peak (nm) |
|---|---|---|---|---|---|---|
| f-ct1a | 4.8 | 14.3, 13.9, 14.3, 12.8 | 23.3, 23.1, 22.9, 20.5 | 0.14, 0.24 | 54 | 476 |
| + v-DABNA | 4.7 | 22.8, 22.0, 22.0, 19.0 | 22.7, 22.4, 22.0, 19.6 | 0.12, 0.14 | 21 | 472 |
| f-ct1b | 3.6 | 14.4, 14.1, 13.8, 12.3 | 23.2, 22.9, 22.3, 19.8 | 0.14, 0.23 | 56 | 476 |
| + v-DABNA | 3.5 | 27.9, 26.9, 24.5, 21.0 | 25.0, 23.9, 22.0, 18.8 | 0.12, 0.12 | 20 | 472 |
| f-ct1c | 2.8 | 20.0, 16.1, 14.4, 11.9 | 30.0, 24.7, 22.0, 16.8 | 0.13, 0.19 | 52 | 472 |
| + v-DABNA | 3.0 | 35.5, 27.1, 24.0, 20.3 | 30.0, 23.4, 21.0, 18.0 | 0.12, 0.11 | 18 | 472 |
| f-ct1d | 2.6 | 11.6, 10.9, 9.7, 7.7 | 20.0, 16.4, 13.9, 10.6 | 0.14, 0.19 | 53 | 472 |
| + v-DABNA | 2.7 | 19.2, 17.9, 15.4, 13.4 | 20.0, 14.9, 13.6, 11.9 | 0.17, 0.12 | 20 | 472 |

[a]Maximum value and data recorded at 100, 1000, and 5000 cd m$^{-2}$, respectively.
[b]Data recorded at 100 cd m$^{-2}$.

host that gives rise to increased exciton-polaron annihilation (EPA). Furthermore, by using 2,8-bis(diphenyl-phosphoryl)-dibenzo[b,d] thiophene (PPT) instead of mCBP host, higher EQEs of 25.2% for f-ct1d and 28.1% for f-ct1c were achieved, showcasing the potential of our emitters for high performance (as shown in Supplementary Fig. 20). It is worth noting that all devices delivered suppressed efficiency roll-offs, which is uncommon for blue phosphorescent OLEDs.

To further explore the potential application of f-ct1a–d, hyper-phosphorescent OLED devices with configurations of ITO/HAT-CN (10 nm)/TAPC (40 nm)/TCTA (10 nm)/mCP (10 nm)/21 wt% f-ct1a–d and 1 wt% v-DABNA in mCBP (20 nm)/TmPyPB (40 nm)/Liq (2 nm)/Al (100 nm) were investigated. Notably, hyperphosphorescence[34], also known as phosphor-sensitized fluorescence[35] or hyper-OLED[36], involves the utilization of a phosphorescent sensitizer to harvest both the singlet and triplet excitons and transfer the energy to terminal emitters via rapid Förster resonance energy transfer (FRET)[37]. Devices employing boron–nitrogen-based TADF emitters such as v-DABNA are capable of taking advantage of both a fast fluorescent lifetime and a narrow emission bandwidth due to the multi-resonance effect[38]. This

process, in accordance with the literature, may solve problems associated with the inferior efficiency and durability of true-blue OLED devices.

Remarkably, our hyper-OLED devices exhibited hyperphosphorescence centered at 472 nm along with a narrow full width at half maximum (FWHM) of 18–21 nm (Fig. 3a). Their emission solely originated from the v-DABNA emitter, suggesting an effective FRET process (Supplementary Figs. 21–23). Thus, in addition to optimal photophysical properties, the six *tert*-butyl groups around the Ir(III) sensitizer are highly effective in separating the v-DABNA terminal emitter and preventing the Dexter energy transfer (DET) process. These devices also show a similar trend in both the turn-on and driving voltages versus their parent devices because the low doping concentration of v-DABNA would not affect charge transport. Above all, high EQEs were observed for hyperphosphorescence. In particular, the f-ct1c-sensitized v-DABNA device not only showed the smallest $y$-value of CIE coordinates (0.12, 0.11) but also exhibited an extraordinarily high EQE$_{max}$ of 35.5% as well as suppressed efficiency roll-off at higher current densities. Figure 3d summarizes the EQE$_{max}$ values of devices

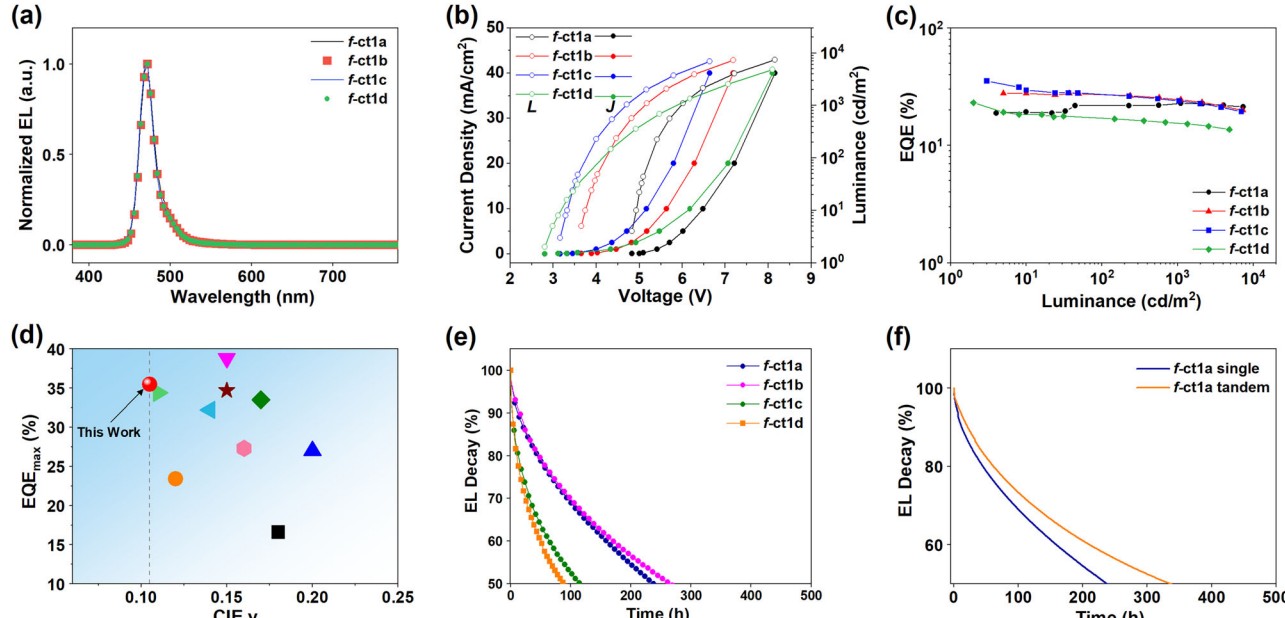

**Fig. 3 | Performances of hyper-OLEDs and operational stabilities.**
**a** Electroluminescence spectra. **b** Current density–voltage–luminance (*J*–*V*–*L*) characteristics. **c** EQE–luminance characteristics of hyperphosphorescent devices using 21 wt% of sensitizers f-ct1a–d. **d** EQE-CIE$_y$ relationship of v-DABNA-based

hyper-OLEDs documented in literature. **e** Luminance vs. operational lifetime of phosphorescent devices using 21 wt% of emitters f-ct1a–d with $L_0 = 500$ cd m$^{-2}$. **f** Normalized luminance as a function of operational time of single and tandem devices using 21 wt% of f-ct1a with $L_0 = 500$ cd m$^{-2}$.

based on v-DABNA with different sensitizers. Clearly, the f-ct1c-sensitized device exhibits unprecedented OLED performance relative to those reported in the current literature (Supplementary Table 4).

Device stability was also estimated using more suitable organic materials and architecture, as mentioned in the "Methods" section. The EL performances of these devices are shown in Supplementary Fig. 25. Our results showed that the lifetimes to 70% of the initial luminance (LT$_{70}$ at $L_0 = 500$ cd m$^{-2}$) for f-ct1a–d were 95, 99, 31, and 25 h (cf., Fig. 3e and Supplementary Fig. 25). The enhancement of the operational lifetimes of f-ct1a and f-ct1b compared to that of the previously reported Ir(cb)$_3$ (LT$_{70} = 52$ h at an initial luminance of 500 cd m$^{-2}$)[39] can be ascribed to the faster radiative transitions of f-ct1a and f-ct1b, which induce accelerated triplet exciton decay. To further improve the device lifetime, sensitized devices with similar architecture (Supplementary Fig. 26) were fabricated, and the device lifetime was greatly enhanced ($T_{70} = 84$ h with an $L_0 = 500$ cd m$^{-2}$). Moreover, we fabricated a tandem OLED (Supplementary Fig. 27) by stacking two EL units with f-ct1a as the emitter and the LiNH$_2$-doped bathophenanthroline (BPhen)/HAT-CN layers as the intermediate connector[40]. The maximum EQE of this tandem OLED was measured to be 20.5%, which is also approximately twice as much as that of the single device. Furthermore, an LT$_{70}$ of 125 h was estimated for the tandem OLED, which is a 100% enhancement versus the single device (Supplementary Fig. 28). Importantly, the LT$_{50}$ value of the tandem device can be predicted to be over 10,000 h at an initial luminance of 100 cd m$^{-2}$ by assuming a lifetime acceleration factor of 1.8. Thus, the phosphorescent and hyper-OLEDs using f-ct1a–d displayed satisfactory stabilities with pure-blue emission.

## Discussion

In conclusion, we synthesized four homoleptic Ir(III)-based carbene emitters f-ct1a– d, in which each imidazo[4,5-b]pyrazinylidene coordination unit was associated with one N-aryl cyclometalate, one vertically arranged N-aryl appendage and two *tert*-butyl substituents at specialized positions. The six *tert*-butyl groups provided the needed steric hindrance to block unwanted stacking interactions between the Ir(III) emitter and the host material in the EML and the terminal emitter

v-DABNA of the hyper-OLED devices. Notably, the three peripheral N-aryl appendages had the face-to-face orientation with respect to their adjacent carbene entity, providing additional spatial protection that was absent in the respective N-alkyl-substituted carbene chelates reported earlier[24,41]. This could be the origin of the observed improvement in device performance. Equilibration of all Ir(III) complexes was unambiguously established with heating in the presence of acidic catalysts (NaOAc + TsOH), with negligible decomposition in high-boiling point chlorinated solvents at ~214 °C. This behavior not only confirmed their high thermal stability but also allowed mass production of needed derivatives, particularly f-ct1b and f-ct1c, for the evaluation of possible industrial applications. Such properties have never been reported in Ir(III)-based carbene emitters, confirming the advantages of the current design. Finally, the OLED device with f-ct1c doped in the EML showed a peak wavelength at 472 nm and a maximum external quantum efficiency (EQE) of 20.0%. Upon introduction of 1 wt% of the terminal emitter v-DABNA, the resulting hyper-OLED exhibited narrowband true-blue hyperphosphorescence with CIE$_{xy}$ coordinates of (0.12, 0.11), substantially reduced FWHM of 18 nm, EQE$_{max}$ of 35.5% and, extraordinarily, retention of high EQE of 20.3% at 5000 cd m$^{-2}$, paving the way for innovative OLED devices.

## Methods

### General information and materials

Commercially available reagents were used without further purification. All solvents were dried and degassed before use, and all reactions were conducted under N$_2$ and monitored using precoated TLC plates (0.20 nm with fluorescent indicator F254). $^1$H NMR spectra were recorded with Bruker 400 MHz "AVANCE III" nuclear magnetic resonance system. Mass spectra were obtained on Bruker microTOF-Q mass spectrometer. TGA measurements were performed on a TA Instrument TGAQ50, at a heating rate of 10 °C min$^{-1}$ under a nitrogen atmosphere.

### Thermal and electrochemical characterization

Thermal gravimetric data were recorded and given in Supplementary Fig. 7 and Supplementary Table 3, to which all complexes display

excellent thermal stability with decomposition temperature ($T_d$) well above 359 °C, among which f-ct1d appeared to possess the highest $T_d$ at 401 °C. Cyclic voltammetry was next measured in acetonitrile at RT (Supplementary Fig. 8 and Supplementary Table 3). All complexes presented reversible oxidation and irreversible reduction waves. The oxidation potentials are mainly occurred at the Ir(III) metal atom and underwent a progressive anodic shift from 0.59, 0.62, 0.67, and 0.69 V vs. Fc$^+$/Fc for f-ct1a–d, respectively. This result agreed with a reduced number of tert-butylphenyl cyclometalate(s). Therefore, f-ct1d is the Ir(III) complex with the most positive oxidation potential according to this deduction. Meanwhile, the optical gaps of 2.84– 2.89 eV were calculated using the emission onset obtained from solution photoluminescence, while their LUMO energy levels were subsequently calculated from the difference of HOMO and optical energy gap recorded.

## Synthesis of all Ir(III) metal complexes (f-ct1a–d)

A degassed o-dichlorobenzene (10 mL) solution of ct1H$_2$·OTf (0.33 g, 0.61 mmol), sodium acetate (82 mg, 1 mmol), and m-IrCl$_3$(tht)$_3$, (0.11 g, 0.2 mmol) was heated at 155 °C for 12 h. After the removal of the solvent, the residue was taken into the CH$_2$Cl$_2$ solution. The organic phase was washed with deionized water, separated, and concentrated to dryness. The residue was separated by column chromatography eluting with n-hexane and ethyl acetate (7/1 to 3/1, v/v), followed by recrystallization. This gave a mixture of four facially arranged Ir(III) complexes, namely: a yellow f-ct1a (20 mg, 7%), a yellow f-ct1b (70 mg, 26%), a light yellow f-ct1c (90 mg, 33%), and a light yellow f-ct1d (24 mg, 9%), respectively. Isomerization was executed by refluxing 1,2,4-trichlorobenzene and in the presence of both p-toluenesulfonic acid and sodium acetate as catalysts.

Spectral data of f-ct1a is provided as follows: MS (ESI, $^{193}$Ir): m/z 1342.6310 [M$^+$]; $^1$H NMR (400 MHz, CDCl$_3$, 296 K) δ 8.67 (d, J = 8.2 Hz, 3H), 8.11 (s, 3H), 7.19 (dd, J = 8.4, 2.0 Hz, 3H), 6.80 (t, J = 7.2 Hz, 3H), 6.76 (d, J = 2.0 Hz, 3H), 6.60 (br, 6H), 1.57 (s, 27H), 1.13 (s, 27H). Anal. calcd. for C$_{75}$H$_{81}$IrN$_{12}$: C, 67.09; H, 6.08; N, 12.52. Found: C, 67.21; H, 6.15; N, 12.38.

Spectral data of f-ct1b is provided as follows: MS (ESI, $^{193}$Ir): m/z 1343.6267 [M$^+$]; $^1$H NMR (400 MHz, CDCl$_3$, 296 K) δ 8.80 (d, J = 8.4 Hz, 1H), 8.73 (d, J = 7.6 Hz, 1H), 8.57 (d, J = 8.0 Hz, 1H), 8.38 (s, 1H), 8.11 (s, 1H), 8.05 (s, 1H), 7.23 (dd, J = 8.0, 2.4 Hz, 1H), 7.17 (td, J = 8.0, 1.6 Hz, 1H), 7.11 (dd, J = 8.0, 2.4 Hz, 1H), 6.84 (td, J = 7.2, 0.8 Hz, 1H), 6.80–6.72 (m, 6H), 6.57 (d, J = 2.0 Hz, 1H), 6.31 (br, 5H), 1.55 (s, 9H), 1.52 (s, 9H), 1.39 (s, 9H), 1.13 (s, 9H), 1.05 (s, 9H), 1.00 (s, 9H). Anal. calcd. for C$_{75}$H$_{81}$IrN$_{12}$: C, 67.09; H, 6.08; N, 12.52. Found: C, 67.25; H, 6.20; N, 12.11.

Spectral data of f-ct1c is provided as follows: MS (ESI, $^{193}$Ir): m/z 1343.6161 [M$^+$]; $^1$H NMR (400 MHz, CDCl$_3$) δ 8.90 (d, J = 8.0 Hz, 1H), 8.69 (d, J = 8.0 Hz, 1H), 8.67 (d, J = 8.0 Hz, 1H), 8.42 (s, 1H), 8.34 (s, 1H), 8.07 (s, 1H), 7.23 (t, J = 7.2 Hz, 1H), 7.19–7.11 (m, 2H), 6.88 (t, J = 7.2 Hz, 1H), 6.82–6.77 (m, 2H), 6.70 (t, J = 7.2 Hz, 1H), 6.62 (d, J = 2.4 Hz, 1H), 6.62 (d, J = 7.2 Hz, 1H), 6.44 (br, 2H), 6.17 (br, 4H), 1.52 (s, 9H), 1.38 (s, 9H), 1.26 (s, 9H), 1.06 (s, 9H), 1.01 (s, 9H), 1.00 (s, 9H). Anal. calcd. for C$_{75}$H$_{81}$IrN$_{12}$: C, 67.09; H, 6.08; N, 12.52. Found: C, 67.20; H, 6.11; N, 12.42.

Spectral data of f-ct1d is provided as follows: MS (ESI, $^{193}$Ir): m/z 1343.6223 [M$^+$]; $^1$H NMR (400 MHz, CDCl$_3$) δ 8.78 (dd, J = 7.6, 0.4 Hz, 3H), 8.39 (s, 3H), 7.18 (td, J = 7.6, 1.2 Hz, 3H), 6.80 (td, J = 7.6, 1.2 Hz, 3H), 6.57 (dd, J = 7.2, 1.2 Hz, 3H), 6.29 (d, J = 8.0 Hz, 6H), 1.26 (s, 27H), 1.03 (s, 27H). Anal. calcd. for C$_{75}$H$_{81}$IrN$_{12}$: C, 67.09; H, 6.08; N, 12.52. Found: C, 67.21; H, 6.14; N, 12.38.

## Computational details of theoretical investigations

The geometries, electronic structures, and electronic excitations of the studied Ir(III) complexes were investigated by methods based on DFT[42,43] and TD-DFT[26,44] using the B3LYP functional[45–47] with Gaussian 16 set of programs[48]. The polarizable continuum model (PCM)[49] was used to include the solvent effects of toluene. The 6–31G(d,p)[50] basis set was used for light elements such as hydrogen, carbon, and nitrogen, while the LANL2DZ[51] basis set with the Los Alamos National Laboratory (LANL) effective core potentials (ECPs) was used for iridium. The corresponding ground state (S$_0$) geometries were optimized based on the X-ray structural data. The low-energy excited states were then calculated by the TD–DFT method based on the optimized ground state structures. Natural transition orbital (NTO)[27] analysis was applied to obtain a clear and compact orbital representation for the excited states described by a variety of orbital transitions without a single predominant one (e.g. T$_1$ state in this work). Avogadro software[52] was used to visualize the orbitals presented in this work (with the isosurface value of the electron density as 0.03 e/au$^3$). Orbital composition analysis was performed using the Hirshfeld method[53] to calculate the contribution of the Ir atom to each molecular orbital with Multiwfn software[54].

## OLED stability

To evaluate the EL stabilities of f-ct1a–d, the OLEDs were fabricated with the following structure: HAT-CN(10 nm)/BCFN (50 nm)/mBCP(5 nm)/mCBP:SiCzTRz:Ir-dopant (30 nm, 21 wt%)/ DBFTRz (5 nm)/ZADN (40 nm)/Liq (3 nm)/Al (100 nm). Chemical structures of BCFN, SiCzTRz, DBFTRz, and ZADN are described in Supplementary Fig. 24. The 1:1 ratio of mCBP:SiCzTRz served as the electroplex co-host[17]. The operational lifetime of devices was recorded by tracing brightness under a constant current. All blue OLEDs show noticeably long lifetime (cf. Fig. 3e and Supplementary Fig. 25). The lifetimes to 70% of the initial luminance (LT$_{70}$ at $L_0$ = 500 cd m$^{-2}$) of devices for f-ct1a–d were 95, 99, 31, and 25 h. Enhancement of the operational lifetime of f-ct1a and f-ct1b compared to that of the previously reported Ir(cb)$_3$ (LT$_{70}$ = 52 h at an initial luminance of 500 cd m$^{-2}$)[19] can be ascribed to the much faster radiative process of f-ct1a and f-ct1b, which induce accelerated triplet exciton decay. The shortened triplet excited-state lifetime of f-ct1a and f-ct1b would reduce the TTA and TPA processes, which allows the pure-blue device to achieve a long lifetime. Noted the magnitude of LT$_{70}$ values of the emitters does not correlate to the ordering of EQEs at 500 cd m$^{-2}$. To better understand the origin of varied operational lifetimes, we further investigated the charge transport behavior in the EML by fabricating hole-only devices (HODs) and electron-only devices (EODs). Supplementary Fig. 29 demonstrated the J–V characteristics of HODs and EODs along with the device structures of unipolar charge devices. For the HODs, devices with f-ct1b and f-ct1c showed a similar current density and voltage rise during operation. On the other hand, for the EODs, f-ct1c device showed a much higher voltage rise than the f-ct1b device. As the f-ct1c device showed a much lower current than the f-ct1b device, the relatively large change in drive voltage for f-ct1c device was attributed to the fact that the f-ct1c caused an electrical trap functioning as a site for long-range hopping transport. We speculated the operational lifetime is still limited by the intrinsic instability of mCBP. Therefore, it is possible to further boost the OLED performance with a long lifetime by employing optimized host materials. After the replacement of mCBP, the device with mCPCN showed similar EL efficiency, together with an outstanding LT$_{70}$ of 135 h (Supplementary Fig. 30).

## Data availability

The data that support our plots and other findings within this paper are available from the corresponding authors upon request. Crystallographic data for the structures reported in this article have been deposited at the Cambridge Crystallographic Data Centre, under deposition numbers CCDC 2170195 (f-ct1a), 2170111 (f-ct1b), 2170110 (f-ct1c) and 2198299 (f-ct1d); copies of the data can be obtained free of charge via https://www.ccdc.cam.ac.uk/structures/.

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

## Acknowledgements
Funding and research grants for this work are provided by the University Grants Council (N_CityU102/19, CityU 11304221, and CityU 11312722), City University of Hong Kong, Hong Kong SAR, Collaborative Innovation Center of Suzhou Nano Science and Technology (CIC-Nano), National Natural Science Foundation of China (61961160731 and 51821002), and National Computational Infrastructure of Australia. X.Z. is a recipient of the Discovery Early Career Researcher Award (ARC DECRA DE190100144) from the Australian Research Council.

## Author contributions
J.Y. and D.-Y.Z. carried out the syntheses and device fabrications and contributed equally to this work. S.-M.Y. executed the single crystal X-ray diffractions. M.K. and X.Z. performed the computational investigations. L.-S.L., X.Z., and Y.C. supervised the projects. All authors discussed the results and contributed to the paper.

## Competing interests
The authors declare no competing interests.
