## [Peer Review File · Nature Communications]

Electroluminescence and hyperphosphorescence from stable blue Ir(III) carbene complexes with suppressed efficiency roll-offREVIEWER COMMENTS

Reviewer #3 (Remarks to the Author):

In this work, the authors designed and synthesized four iridium complexes, which resemble Ir(cb)₃, in my opinion. The authors also investigated the isomerization of the transformation of the four isomers at elevated temperatures. The authors reported a high EQEmax of 35.5% for OLEDs with f-ct1c as the sensitizer and v-DABNA as the final emitter. Major revisions are needed before consideration of publication in nature communications:

1. The molecular design shows of insufficient novelty. f-ct1a has one tBu group on the phenyl of Ir(cb)₃ and the other molecules are isomers of f-ct1a.
2. The device performance of the phosphorescent OLEDs are moderate, with the highest EQEmax of 20% for f-ct1c, lower than Ir(cb)₃ of 24.8%, CIEy 0.137 in 2018. (Adv. Funct. Mater. 2018, 28, 1802945).
3. The stable device has a different device structure, and only the device lifetime is provided without EQE vs. luminance and J-V-L spectra
4. The hyper OLED performance looks good, but device stability was not tested.
5. The paper did not provide enough discussions on the influence of the isomers, which I think should be the key point of the manuscript.
6. Figure 2 shows the J-V-L characteristics of devices, however, it did not show the starting voltage from low luminance. Please provide the original data, as the turn-on voltage is defined as the voltage for 1 cd m⁻².
7. The turn-on voltages of these four isomers range from 2.6 V to 4.8 V, with big differences. The authors contributed this to the hole traps. However, the HOMO levels of these four complexes are -5.39 eV, -5.42 eV, -5.47 eV, and -5.49 eV, respectively. No significant difference. Please explain it with more strong evidence.
8. In Figure S10 to Figure S14, first, the EQEmax generally increases as the concentration of the dopant increases for f-ct1a-c. However, for f-ct1d, when the concentration increases from 7% to 14% and 21%, the EQEmax is reduced gradually. Please explain the reason.
9. Please explain the difference of k_r between the isomers theoretically, especially for f-ct1d.
10. In line 172 (Moreover, their EL spectra showed minimal broadening in comparison to the PL spectra). However, the EL spectra of the phosphorescent OLEDs in Table 2 are narrower than the PL spectra in Table 1. Why?
11. In the synthesis section, Line 284 wrote (1H and 19F NMR spectra were recorded with Bruker 400 MHz "AVANCE III"). However, there is no F in the structure.

12. There are some errors and typos in the manuscript (e.g., SI Figure S9, the word toluene should be toluene.)

Reviewer #4 (Remarks to the Author):

As the emitters for the practical OLEDs, the efficiency of blue fluorescence emitters are much lower than the green and red Ir(III) complexes. Therefore, the development of the blue phosphorescent and TADF materials are very important. In this paper, the authors reported four Ir(III) metal complexes bearing functionalized imidazo[4,5-b]pyrazinylidene fragments, which exhibit efficient blue emissions and a fast radiative decay lifetime. The OLED with one Ir(III) complex shows a peak wavelength at 472 nm and an EQE of 20.0%. Furthermore, using the Ir(III) complex as the phosphorescent sensitizer via Förster energy transfer to one MR-TADF material v-DABNA, the hyper-OLED exhibits a true-blue color, a FWHM of 18 nm, a maximum EQE of 35.5% and a high EQE of 20.3% at 5000 cd m⁻². And the device also shows relative long operation lifetime by compared with the previously reported Ir(cb)₃. I think this paper shows some important results for achieving superb blue emissive OLEDs. But the authors must pay attention to the following points.

1. The PL peaks of four Ir(III) complexes are in the required range around 465 nm with high efficiencies. But all devices presented blue emissions with peak wavelengths spanning a range from 472 to 476 nm with the mCBP host. Have you tried other hosts to maintain the emission peaks around 465 nm?

2. Four Ir(III) complexes have similar photophysical properties, but why the performances of their devices show so great difference? Furthermore, the f-ct1c based device delivered the highest EQEmax of 20.0%, why the authors described the device performances of f-ct1d in detail? From my opinion, the device with f-ct1d emitter should show the much serious efficiency roll-off due to the longest lifetime. Furthermore, I also noticed the dopant concentration is as high as 21 wt%.

3. The MR-TADF materials always show high device efficiencies because their high PLQY, but the efficiency roll-offs are serious due to the much long triplet state lifetimes. How about the lifetime of the doped films of 21 wt% f-ct1a – d and 1 wt% v-DABNA in mCBP.

Reviewer #5 (Remarks to the Author):

In this work, the authors synthesized four blue emitting Ir(III) carbene complexes f-ct1a-d, and a max. EQE of 33.5% and electroluminescence peak of 472 nm have been achieved with f-ct1c as sensitizer and v-DABNA as emitter, which expands the application of phosphorescent materials in the field of blue OLED. So, I think it can be recommended for publication after minor revision. The following questions are raised for reference.

1. In the analysis of Figure S15, the authors believe that, "a higher dopant concentration also leads to an enhancement in the electron current." However, from Figure S15(d), it appears that the electron current decreases with increasing doping concentration. Please provide an explanation.
2. The phosphorescent device based on f-ct1a exhibits a higher turn-on voltage and lower external quantum efficiency. The authors attribute this phenomenon to the increased HOMO gap between the dopant and host that gives rise to increased exciton-polaron annihilation (TPA). However, the sensitization device using f-ct1a as the sensitizer and v-DABNA as the emitter exhibits the longest device lifetime, which seems to conflict with the previous discussion. Please provide an explanation.
3. Given the good stability of the f-ct1a sensitized v-DABNA devices, it is better to supply transient EL analysis, similar to the transient PL traces shown in Figure S18.

Dear reviewers:

We are very grateful to all reviewers for the useful comments, so that we can have the opportunity to improve the contents of this contribution. Now, we have revised the manuscript according to the comments and our point-by-point responses are listed below:

Replies to the Reviewers (reviewer's comments are in blue, and the replies are in black)

Reviewer#3

In this work, the authors designed and synthesized four iridium complexes, which resemble Ir(cb)₃, in my opinion. The authors also investigated the isomerization of the transformation of the four isomers at elevated temperatures. The authors reported a high EQEmax of 35.5% for OLEDs with f-ct1c as the sensitizer and v-DABNA as the final emitter. Major revisions are needed before consideration of publication in nature communications:

Reply: We are very grateful to the positive comments made on our manuscript as well as the recommendation to have this work published in Nat. Commun. after necessary revisions. The point-by-point corrections are listed below.

1. The molecular design shows of insufficient novelty. f-ct1a has one tBu group on the phenyl of Ir(cb)₃ and the other molecules are isomers of f-ct1a.

Reply: Indeed, the difference between Ir(cb)₃ and our samples was the single t-butyl substituent selectively attached to one of the N-phenyl groups. However, with introduction of this t-butyl group, we are able to solve the long lasting problem on separation of isomeric products for this class of Ir(III) carbene emitters and to confirm the fast equilibration and interconversion among isomers in presence of catalyst at higher temperature. This is the impact of our molecular designs. To further emphasize this achievement, we added a statement in page 5 of revised manuscript, under the session of Catalytic interconversion in solution. These statements are: "*The chemistry of these Ir(III) emitters f-ct1a-d is remarkable. After introduction of a t-butyl group to one N-aryl group of Ir(cb)₃, in giving isomeric f-ct1a – d, we can solve the problem on product separation of Ir(cb)₃ analogues and to confirm the fast equilibration and interconversion among all available isomers f-ct1a – d in presence of catalyst at higher temperature.*"

2. The device performance of the phosphorescent OLEDs are moderate, with the highest EQEmax of 20% for f-ct1c lower than Ir(cb)₃ of 24.8%, CIEy 0.137 in 2018. (Adv. Funct. Mater. 2018, 28, 1802945).

Reply: We are very grateful to the insightful comment. We have carefully reviewed the reference paper (Adv. Funct. Mater. 2018, 28, 1802945), which employed a new bipolar material, m-CBPPO, as an effective host for emitter Ir(cb)₃, leading to an improved maximum EQE of 24.8%. The reference highlighted the critical roles of a host material in determining the EL performance of the dopant.

To enhance the EL performance of our emitters, we have also chosen a similar 2,8-bis(diphenyl-phosphoryl)-dibenzo[*b,d*]thiophene (PPT) as the host material during device fabrication. As a result, all emitter's exhibit improved EL efficiency compared to the cases using mCBP as the host material. Notably, our devices achieve a higher EQE of 28.1% for **f-ct1c** and 25.2% for **f-ct1d**, as depicted in **Figure R1** (or **Supplementary Fig. 20**). However, it is important to acknowledge that PPT has a reputation for instability, despite its usually reported high EQE when employed as the host material for phosphorescent and TADF emitters. In our current work, we prefer to emphasize the improvements in stability, and therefore, we did not include this data in the main manuscript. However, in response to this comment, we have incorporated the EL data based on PPT in the supporting information to showcase the potential of our emitters for achieving high performance. Furthermore, we have included relevant descriptions in our revised manuscript (on Page 8) to address this aspect.

*“Furthermore, by using 2,8-bis(diphenyl-phosphoryl)-dibenzo[*b,d*]thiophene (PPT) instead of mCBP as host, higher EQEs of 25.2% for **f-ct1d** and 28.1% for **f-ct1c** were achieved, showcasing the potential of our emitters for high performance (as shown in Supplementary Fig. 20.”*

Figure R1 (or Supplementary Fig. 20). (a) Schematic diagram of the OLED devices; (b) Molecular structure of PPT; (c) Current density-voltage-luminance (J-V-L) characteristics; (d) EL spectra; (e) CE-luminance characteristics; (f) EQE-luminance characteristics. All doping conc. of Ir(III) complexes is maintained at 21 wt%.

3. The stable device has a different device structure, and only the device lifetime is provided without EQE vs. luminance and J-V-L spectra

Reply: We are very thankful to this comment. We regret not to mention the EL performance of the stable devices in the main text. However, we would like to clarify that the J-V-L and EQE vs. luminance characteristics have indeed been included in

Figure R2 (or Supplementary Fig. 25) of the supporting information. In response to this comment, we have updated the figures of device performance and added the description of the relevant data on Page 9.

“The EL performances of these devices are shown in **Supplementary Fig. S25.**”

Figure R2 (or Supplementary Fig. 25). (a) Schematic device structure; (b) Current density-voltage-luminance (J-V-L) characteristics; (c) EL spectra; (d) CE-luminance characteristics; (e) EQE-luminance characteristics; (f) Normalized luminance with a $L_0 = 500 \text{ cd m}^{-2}$ as a function of operational time. All doping conc. of Ir(III) complexes is maintained at 21wt%.

4. The hyper OLED performance looks good, but device stability was not tested.

Reply: Thank you for your positive evaluation on our device results. In response to this comment, we have measured the stability of the hyper-OLEDs with a structure of “ITO/HAT-CN (10 nm)/TAPC (40 nm)/TCTA (10 nm)/mCP (10 nm)/mCBP:21 wt% **f-ct1c**:1 wt% v-DABNA (20 nm)/TmPyPB (40 nm)/Liq (2 nm)/Al (100 nm)”. However, due to the inadequate intrinsic instability of TAPC, mCBP, and TmPyPB materials, this OLED device delivers an inferior lifetime ($T_{70} = 24 \text{ h}$ with a $L_0 = 500 \text{ cd m}^{-2}$), which is consistent with results reported in the reference (Nam *et al. Adv. Sci.* **2021**, *8*, 2100586). To further improve the device lifetime, we fabricated a new device having a similar structure with our stable phosphorescent OLEDs: “ITO/HAT-CN (10 nm)/BCFN (40 nm)/mCBP (10 nm)/mCBP:SiCzTRz:21 wt% **f-ct1c**:1 wt% v-DABNA (20 nm)/DFRTRz (10 nm)/ZADN (40 nm)/Liq (2 nm)/Al (100 nm)”. The lifetime data of the device is shown in **Figure R3 (or Supplementary Fig. 26)**. Notably, the device lifetime is greatly improved ($T_{70} = 84 \text{ h}$ with a $L_0 = 500 \text{ cd m}^{-2}$), which is comparable to the sensitized device with v-DABNA. In response to this comment, we have added the description on the stability of the sensitized devices, see highlighted text in page 10 of the main manuscript.

“To further improve the device lifetime, new sensitized devices with similar architecture (**Supplementary Fig. 26**) were fabricated, and the device lifetime was greatly enhanced ($T_{70} = 84$ h with a $L_0 = 500$ cd m^{-2}).”

Figure R3 (or Supplementary Fig. 26). Sensitized OLEDs based on mCBP and mCBP:SiCzTRz as the host. (a) Schematic device structures; (b) Current density-voltage-luminance (J-V-L) characteristics; (c) EL spectra, (d) CE-luminance characteristics; (e) EQE-luminance characteristics; (f) Normalized luminance with a $L_0 = 500$ cd m^{-2} as a function of operational time. All doping conc. of Ir(III) complexes is maintained at 21 wt%.

5. The paper did not provide enough discussions on the influence of the isomers, which I think should be the key point of the manuscript.

Reply: To address these points, we extended the discussion at the end of Section “Photophysical properties and theoretical studies”, c.f., see highlighted text in page 5. The newly added descriptions are: “*Their interplay leads to an enhanced radiative rate constant (k_r) for **f-ct1c**, as a higher MLCT percentage in the T_1 excited states should give a faster k_r due to the direct involvement of metal atoms that facilitate phosphorescence, and the increasing orbital overlap also results in an increased k_r due to an increased transition dipole moment. It is also noted that **f-ct1d** achieves the highest PLQY among all four isomers with both lowered k_r and k_{nr} . This might be due to its relative enhanced structural stability leading to a relative higher activation barrier from its emissive state to the non-emissive metal-centered states and thus a lower k_{nr} .*”

6. Figure 2 shows the J-V-L characteristics of devices, however, it did not show the starting voltage from low luminance. Please provide the original data, as the turn-on voltage is defined as the voltage for 1 cd m^{-2} .

Reply: We appreciate the reviewer’s comment. The value of V_{on} was recorded by extracting the L-V curves to the 1 cd m^{-2} . To address this concern and demonstrate the reliability of our extraction method, we conducted additional device fabrication based on **f-ct1c** and measurements starting from lower current densities ($0.005 \text{ mA}\cdot\text{cm}^{-2}$). The results of these experiments are depicted in **Figure R4** and **Table R1**. Upon examination, it is evident that the turn-on voltage (V_{on}) values obtained directly from the luminance-voltage (L-V) curves and those extracted using our method are identical. This consistency in results provides evidence supporting the trustworthiness of the V_{on} values reported in our work.

Figure R4. Luminance versus voltage characteristics of OLEDs measured from different initial current density (dot: $0.01 \text{ mA}\cdot\text{cm}^{-2}$; dash line: $0.005 \text{ mA}\cdot\text{cm}^{-2}$).

Table R1. Original data of the OLED measured from different starting current density.

Sample 1			Sample 2		
Current Density (mA cm^{-2})	Voltage (V)	Luminance (cd m^2)	Current Density (mA cm^{-2})	Voltage (V)	Luminance (cd m^2)
0.01	2.84714	3	0.005	2.77498	1.185
0.02	2.93043	5	0.01	2.84067	2.44
0.05	3.08949	13	0.015	2.88661	3.669
0.1	3.2612	26	0.02	2.9237	4.895
0.2	3.47151	51	0.05	3.07377	12.26
0.5	3.81579	125	0.1	3.24017	24.42
1	4.1307	245	0.5	3.78139	118
2	4.5094	477	1	4.08372	229.7
5	5.15647	1137	5	5.04458	1059
10	5.8425	2137	10	5.64822	1985
20	6.73403	3890	20	6.46766	3613
40	7.8008	6838	40	7.47921	6365
60	8.48692	9240	60	8.11556	8613
80	9.01767	11280	80	8.59082	10490

7. The turn-on voltages of these four isomers range from 2.6 V to 4.8 V, with big differences. The authors contributed this to the hole traps. However, the HOMO levels of these four complexes are -5.39 eV, -5.42 eV, -5.47 eV, and -5.49 eV, respectively. No significant difference. Please explain it with more strong evidence.

Reply: We are very grateful to this valuable comment. To verify the EL data, we have conducted a new batch of OLED fabrication. The J-V-L characteristics of OLEDs in **Figure R5** shows the same trend of turn-on voltage as the data in **Figure 2b**: **f-ct1a** > **f-ct1b** > **f-ct1d** > **f-ct1c**. The reproducibility of the EL data indicates the differences in turn-on voltage are reasonable, which are not caused by the variations between batch-to-batch experiments.

Figure R5. Current density-luminance-voltage characteristics of OLEDs based on **f-ct1a – 1d** as the emitters. Doping conc. of all Ir(III) complexes is maintained at 21 wt%.

To verify the effects of energy gap between HOMOs of the host and guest, we fabricated the hole-only devices (HODs) with the structure of “ITO/HAT-CN(10 nm)/TAPC(30 nm)/TCTA(10 nm)/mCP(10 nm)/mCBP:21% **f-ct1a – 1d** (20 nm)/mCP(10 nm)/TCTA(10 nm)/TAPC(10 nm)/HAT-CN(10 nm)/Al(80 nm)”. **Figure R6** depicts the HODs composing different EMLs showing clearly different J-V behaviors. For **f-ct1a** and **f-ct1b**, the trap-filled limited regions appear in the J-V curves, while for **f-ct1c** and **f-ct1d**, the J-V curves show a slow transition from Ohmic region to space charge limited current region.

Figure R6. Current density versus voltage characteristics of hole-only devices based on **f-ct1a – 1d** as the emitters. All doping conc. of all Ir(III) complexes is maintained at 21 wt%.

Furthermore, we have fabricated a set of OLEDs using TAPC as the host (ITO/HAT-CN(10 nm)/TAPC(40 nm)/TCTA(10 nm)/mCP(10 nm)/TAPC:21% **f-ct1a** or **f-ct1c** (20 nm)/TmPyPB(40 nm)/Liq(2 nm)/Al(80 nm)) or thinner EMLs (ITO/HAT-CN(10 nm)/TAPC(40 nm)/TCTA(10 nm)/mCP(10 nm)/mCBP:21% **f-ct1a** or **f-ct1c** (5 nm)/TmPyPB(40 nm)/Liq(2 nm)/Al(80 nm)). Compared to mCBP (ca. -6.1 eV), TAPC has a much lower-lying HOMO of ca. -5.5 eV, which could result in less mismatch of the HOMO energy levels. As shown in **Figure R7a**, the device with **f-ct1a**-dope TAPC as EML exhibits a much lower turn-on voltage than the mCBP device. The results are also consistent with previous work (Sanderson *et al. Appl. Phys. Lett.* **2019**, *115*, 263301; Cheng *et al. Sci. Rep.* **2019**, *9*, 3654). The sensitivity of turn-on voltage with the thickness of trap containing EML (**Figure R7b**) further confirm the traps are present due to the energy gap between HOMOs of the hosts and the guests. Based on the above results, we proposed the charge transport mechanism of the EMLs shown in **Figure R8**. For reducing the overall length of this article, we decided not to include the associated experiments into our revised manuscript nor the supplementary

information. The interested readers can access this reviewer's comments and our replies in accordance with the publication policy of *Nat. Commun.*

Figure R7. Current density versus voltage characteristics of OLEDs based on (a) different hosts and (b) different EML thicknesses. All doping conc. of Ir(III) complexes is maintained at 21wt%.

Figure R8. Schematic illustration of charge transport and recombination processes in different EML based on (a) **f-ct1a** and (b) **f-ct1c**. All doping conc. of Ir(III) complexes is maintained at 21wt%.

8. In Figure S10 to Figure S14 (Figure S10 to Figure S14 have been changed to **Supplementary Fig. 12** to **Supplementary Fig. 15** in the revised supplementary information), first, the EQEmax generally increases as the concentration of the dopant increases for f-ct1a-c. However, for f-ct1d, when the concentration increases from 7% to 14% and 21%, the EQEmax is reduced gradually. Please explain the reason.

Reply: We appreciate this comment. The increase in dopant concentration generally leads to the generation of more excitons on the phosphors, facilitated by the Dexter energy transfer from the host to the dopant. **f-ct1d**, however, it exhibits the longest exciton lifetime among the emitters, as evidenced by the transient PL traces in **Supplementary Fig. 19**. The slow decay rate associated with **f-ct1d** can result in the dominance of processes such as triplet-triplet annihilation (TTA) or two-photon

absorption (TPA). These processes, while reducing the exciton quenching, can also contribute to a decrease in the electroluminescent (EL) efficiency.

9. Please explain the difference of k_r between the isomers theoretically, especially for **f-ct1d**.

Reply: There exists a steady decrease of electron donating t-butyl substituents on the phenyl cyclometalates from **f-ct1a** to **f-ct1d**. It then reduced the relative electron density at the Ir(III) metal center, to which the reduced MLCT contribution will result in the decreased k_r for **f-ct1d** under the typical situation. However, our calculation also showed that **f-ct1d** achieves the highest PLQY among all four isomers with both lowered k_r and k_{nr} . This might be due to its relative enhanced structural stability leading to a relative higher activation barrier from its emissive state to the non-emissive metal-centered states and thus a lower k_{nr} . Hence, we now added the following statement: *“Their interplay leads to an enhanced radiative rate constant (k_r) for **f-ct1c**, as a higher MLCT percentage in the T_1 excited states should give a faster k_r due to the direct involvement of metal atoms that facilitate phosphorescence, and the increasing orbital overlap also results in an increased k_r due to an increased transition dipole moment. It is also noted that **f-ct1d** achieves the highest PLQY among all four isomers with both lowered k_r and k_{nr} . This might be due to its relative enhanced structural stability leading to a relative higher activation barrier from its emissive state to the non-emissive metal-centered states and thus a lower k_{nr} .”* They were inserted into the highlighted statement in page 5.

10. In line 172 (Moreover, their EL spectra showed minimal broadening in comparison to the PL spectra). However, the EL spectra of the phosphorescent OLEDs in Table 2 are narrower than the PL spectra in Table 1. Why?

Reply: We believed that such change in FWHM is due to the media of measurement. In general, PL was measured in toluene at RT, which typically exhibited greater solvatochromic effect than that in the co-doped host material for EL measurement.

11. In the synthesis section, Line 284 wrote (1H and 19F NMR spectra were recorded with Bruker 400 MHz "AVANCE III"). However, there is no F in the structure.

Reply: We are grateful to the reminder and this typo is now deleted.

12. There are some errors and typos in the manuscript (e.g., SI Figure S9, the word toluene should be toluene.)

Reply: Again, we are very grateful to the reminder and this typo is now corrected.

Reviewer#4

As the emitters for the practical OLEDs, the efficiency of blue fluorescence emitters are much lower than the green and red Ir(III) complexes. Therefore, the development of the blue phosphorescent and TADF materials are very important. In this paper, the authors reported four Ir(III) metal complexes bearing functionalized imidazo[4,5-

b]pyrazinylidene fragments, which exhibit efficient blue emissions and a fast radiative decay lifetime. The OLED with one Ir(III) complex shows a peak wavelength at 472 nm and an EQE of 20.0%. Furthermore, using the Ir(III) complex as the phosphorescent sensitizer via Förster energy transfer to one MR-TADF material v-DABNA, the hyper-OLED exhibits a true-blue color, a FWHM of 18 nm, a maximum EQE of 35.5% and a high EQE of 20.3% at 5000 cd m⁻². And the device also shows relative long operation lifetime by compared with the previously reported Ir(cb)₃. I think this paper shows some important results for achieving superb blue emissive OLEDs. But the authors must pay attention to the following points.

Reply: We are very grateful for the positive comments made on our manuscript as well as the recommendation to have this work published in Nat. Commun. after necessary revisions. The point-by-point corrections are listed below.

1. The PL peaks of four Ir(III) complexes are in the required range around 465 nm with high efficiencies. But all devices presented blue emissions with peak wavelengths spanning a range from 472 to 476 nm with the mCBP host. Have you tried other hosts to maintain the emission peaks around 465 nm?

Reply: We appreciate the reviewer's comment. The differences between EL and PL spectra could be attributed to a few of reasons, including the status of materials, polarity of the matrix, optical cavity effects, doping concentration, and testing equipment, etc. In our main text on Page 6, we have provided an explanation on the redshift of EL spectra relative to PL spectra, attributing it to the change in polarity resulting from the solvent and host matrix. To explore the possibility of maintaining the EL emission around, we have conducted additional experiments using different hosts, such as mCPCN, SiCzTRZ, PPT, PPCzTRz, and CBP, with their molecular structures shown in **Figure R9**. **Figure R9** presents the EL spectra of these devices, which exhibit a red-shift trend in the order of mCPCN, mCBP, SiCzTRZ, PPT, PPCzTRz, and CBP. However, the EL emission peaks remain located at 472 nm, although mCPCN and mCBP show the bluest emissions.

On the other hand, redshifts of the EL emission relative to the PL emission typically occur for OLED emitters (Zhang *et al.* *ACS Appl. Mater. Interfaces* **2022**, *14*, 1546–1556; Chen *et al.* *Adv. Optical Mater.* **2022**, *10*, 2101952), which can be attributed to the optical cavity effects of OLED devices. To confirm this hypothesis, we fabricated a series of OLEDs with different thicknesses of the electron transport layer (ETL). The EL spectra based on different ETL thicknesses are shown in **Figure R9**. When the ETL thickness is 20 nm, the emission peak shifts to 464 nm (note that the resolution of Photoresearch 650 Instrument used for EL measurements is 4 nm), which is close to the value measured in the PL spectra.

To make a clear demonstration of the EL spectra, we have modified the sentence in the revised manuscript:

*“The EL peak wavelengths of **f-ct1c** and **f-ct1d** are blue shifted in reference to those of **f-ct1a** and **f-ct1b**, while all EL peaks are found to be red shifted by ~8 nm relative to the corresponding photoluminescence (PL) spectra (**Supplementary Fig. 11**), which could be attributed to the change of polarity of solvent and host matrix and the*

optical microcavity effects within OLED devices (**Supplementary Fig. 12 – 15**).” This statement is shown as the highlighted text in page 6 of our revised manuscript.

Figure R9. OLEDs use **f-ct1c** as the emitter. (a) Molecular structures of hosts; (b) EL spectra of OLED based on different host; (c) EL spectra of OLED based on different ETL thicknesses for mCBP.

2. Four Ir(III) complexes have similar photophysical properties, but why the performances if their devices show so great difference? Furthermore, the f-ct1c based device delivered the highest EQEmax of 20.0%, why the authors described the device performances of f-ct1d in detail? Form my opinion, the device with f-ct1d emitter should show the much serious efficiency roll-off due to the longest lifetime. Furthermore, I also noticed the dopant concentration is as high as 21 wt%.

Reply: We are very grateful to this comment. Although similar photophysical properties have been demonstrated, the four emitters exhibit considerably different charge transport properties (**Figure R10**) due to their different arrangement of chelates, which may influence the carrier balance and recombination and consequently the EL performances.

As to the second question, after carefully checking the main text, we have not found any discussion on the OLED device regarding the emitter **f-ct1d**. However, we do agree the device with **f-ct1d** emitter shows the much inferior efficiency roll-off due to the longest radiative lifetime, as shown in **Table 1** and **Figure 2c** and **2d**.

Figure R10. Current density versus voltage characteristics of (a) hole-only devices and (b) electron-only devices based on **f-ct1a – 1d** as the emitters. Doping conc. of Ir(III) complexes is maintained at 21wt%.

3. The MR-TADF materials always show high device efficiencies because their high PLQY, but the efficiency roll-offs are serious due to the much long triplet state lifetimes. How about the lifetime of the doped films of 21 wt% f-ct1a – d and 1 wt% v-DABNA in mCBP.

Reply: We are very grateful to this valuable comment. In response to this comment, the lifetime of the doped films of 21 wt% **f-ct1a – d** and 1 wt% v-DABNA in mCBP have been measured. As shown in **Figure R11**, the comparable PL lifetime indicates that the delayed fluorescence of the MR-TADF emitter is remarkably suppressed due to the efficient energy transfer from the sensitizers to the v-DABNA. As a result, low efficiency roll-off has been achieved by using the phosphors as the sensitizer in the hyperphosphorescence devices.

Figure R11. Time-resolved PL intensity of mCBP:**f-ct1a – d**:v-DABNA films.

Reviewer#5

In this work, the authors synthesized four blue emitting Ir(III) carbene complexes **f-ct1a-d**, and a max. EQE of 33.5% and electroluminescence peak of 472 nm have been achieved with **f-ct1c** as sensitizer and **v-DABNA** as emitter, which expands the application of phosphorescent materials in the field of blue OLED. So, I think it can be recommended for publication after minor revision. The following questions are raised for reference.

Reply: We are very grateful to the positive comments made on our manuscript as well as the recommendation to have this work published in Nat. Commun. after necessary revisions. The point-by-point corrections are listed below.

1. In the analysis of Figure S15 (Figure S15 has been changed to **Supplementary Fig. 17** in the revised supplementary information), the authors believe that, “a higher dopant concentration also leads to an enhancement in the electron current.” However, from Figure S15(**d**), it appears that the electron current decreases with increasing doping concentration. Please provide an explanation.

Reply: Thanks for the kind comment. After checking on the original data and Figure S15, we find the electron current indeed decreases with the increasing doping ratio. We are sorry for the mistake. In response to this comment, we have corrected this sentence in the revised manuscript, see highlighted text on page 8.

2. The phosphorescent device based on **f-ct1a** exhibits a higher turn-on voltage and lower external quantum efficiency. The authors attribute this phenomenon to the increased HOMO gap between the dopant and host that gives rise to increased exciton-polaron annihilation (TPA). However, the sensitized device using **f-ct1a** as the sensitizer and **v-DABNA** as the terminal emitter exhibits the longest device lifetime, which seems to conflict with the previous discussion. Please provide an explanation.

Reply: We are very grateful to this comment. We would like to clarify that the long lifetimes were achieved in the phosphorescent device without the v-DABNA emitter (**Figure 3e**). Due to the comparable lifetime of the **f-ct1a** and **f-ct1b** devices, we have used the **f-ct1b** to fabricate the hole-only and electron-only devices and conducted the electrical aging test on them to investigate the degradation mechanism of OLEDs. Compared to the **f-ct1c**, the **f-ct1b** devices exhibit a much smaller voltage increase, indicative of the formation of less quenchers during the electron current flows. As a result, the **f-ct1b** device exhibits a longer lifetime than the **f-ct1c** device.

3. Given the good stability of the f-ct1a sensitized v-DABNA devices, it is better to supply transient EL analysis, similar to the transient PL traces shown in Figure S18 (Figure S18 has been changed to **Supplementary Fig. 21** in the revised supplementary information).

Reply: We are very grateful to this valuable suggestion. It is worth to clarify that the good stability reported in this work was based on a **f-ct1a** phosphorescent OLED without sensitization. The transient PL traces in **Supplementary Fig. 21** well demonstrates the energy transfer paths in the sensitization system. To further explore the EL mechanism in the **f-ct1a** sensitized device, we have fabricated three OLEDs, which have a general structure of “ITO/HAT-CN (10 nm)/TAPC (40 nm)/TCTA (10 nm)/mCP (10 nm)/EML (20 nm)/TmPyPB (40 nm)/Liq (2 nm)/Al (100 nm)”, in which the EML is composed of mCBP:21 wt% **f-ct1a**, mCBP:1 wt% v-DABNA, and mCBP:21 wt% **f-ct1a**:1 wt% v-DABNA, respectively. Although the slow EL decay rates are observed in the **f-ct1a** devices (**Figure R12**), the result yet cannot give a direct relationship between the device stability.

Figure R12. Transient EL traces of OLEDs based on mCBP:21 wt% **f-ct1a**, mCBP:1 wt% v-DABNA, and mCBP:21 wt% **f-ct1a**:1 wt% v-DABNA as the EML.

REVIEWERS' COMMENTS

Reviewer #3 (Remarks to the Author):

The manuscript is acceptable as is.

Reviewer #4 (Remarks to the Author):

I think the authors revised their paper according to our comments, and it can be accepted for publication in present form.

Reviewer #5 (Remarks to the Author):

All my concerns are addressed in the revised manuscript. It can be recommended for publication as is.